# Human assumed central sensitisation (HACS) in patients with chronic low back pain radiating to the leg (CLaSSICO study)

Ingrid Schuttert [ID],[1] Hans Timmerman [ID],[1] Gerbrand J Groen [ID],[1] Kristian Kjær Petersen [ID],[2] Lars Arendt-Nielsen [ID],[2,3] Andre P Wolff [ID] [1]

¹Department of Anesthesiology, Pain Center, University Medical Centre Groningen, Groningen, The Netherlands
²Department of Health Science and Technology, Aalborg Universitet, Aalborg, Denmark
³Department of Medical Gastroenterology (Mech-Sense), Aalborg University Hospital, Aalborg, Denmark

**Correspondence to**
Ingrid Schuttert;
i.schuttert@umcg.nl

## ABSTRACT

**Introduction** Patients with chronic low back pain radiating to the leg (CLBPr) are sometimes referred to a specialised pain clinic for a precise diagnosis based, for example, on a diagnostic selective nerve root block. Possible interventions are therapeutic selective nerve root block or pulsed radiofrequency. Central pain sensitisation is not directly assessable in humans and therefore the term 'human assumed central sensitisation' (HACS) is proposed. The possible existence and degree of sensitisation associated with pain mechanisms assumed present in the human central nervous system, its role in the chronification of pain and its interaction with diagnostic and therapeutic interventions are largely unknown in patients with CLBPr. The aim of quantitative sensory testing (QST) is to estimate quantitatively the presence of HACS and accumulating evidence suggest that a subset of patients with CLBPr have facilitated responses to a range of QST tests.

The aims of this study are to identify HACS in patients with CLBPr, to determine associations with the effect of selective nerve root blocks and compare outcomes of HACS in patients to healthy volunteers.

**Methods and analysis** A prospective observational study including 50 patients with CLBPr. Measurements are performed before diagnostic and therapeutic nerve root block interventions and at 4 weeks follow-up. Data from patients will be compared with those of 50 sex-matched and age-matched healthy volunteers. The primary study parameters are the outcomes of QST and the Central Sensitisation Inventory. Statistical analyses to be performed will be analysis of variance.

**Ethics and dissemination** The Medical Research Ethics Committee of the University Medical Center Groningen, Groningen, the Netherlands, approved this study (dossier NL60439.042.17). The results will be disseminated via publications in peer-reviewed journals and at conferences.

**Trial registration number** NTR NL6765.

## INTRODUCTION
### Background

The mean incidence and prevalence of chronic low back pain radiating to the leg (CLBPr) are 9.4 and 17.2 per 1000 person years, respectively.[1] If the diagnosis of

## Strengths and limitations of this study

► Comparison with sex-matched and age-matched healthy controls.
► Multimodal assessment of human assumed central and peripheral sensitisation via objective and subjective measures.
► The order of the quantitative sensory testing measures is not randomised for each patient and each visit and is only randomised per visit.

CLBPr is unclear despite extensive neurological, orthopaedic and radiological examination, patients are sometimes referred to a specialised multidisciplinary pain clinic, where a diagnostic selective nerve root block (dSNRB) might provide a more precise diagnosis.[2 3] This block, achieved by injection of a local anaesthetic, can help determine the predominant segmental level of the pain.[4–9] A positive dSNRB is generally followed by an injection of local anaesthetics and corticosteroids, that is, a therapeutic SNRB (tSNRB) or a pulsed radiofrequency treatment (pRF) of the spinal nerve root, its dorsal ganglion or a combination thereof.

Growing evidence suggests that central pain mechanisms are facilitated in a subset of patients with CLBPr.[10 11] Facilitation of peripheral and central pain mechanisms have been suggested to be associated with pain progression, poor recovery, poor response to pharmacological interventions and inadequate response to low back surgery, indicating the clinical need of these assessments.[12–14] Preclinical data suggest that nerve fibres can malfunction and respond as nociceptors, resulting in pain from light stimuli, possibly caused by peripheral and central sensitisation.[15] Dysfunction of the somatosensory system may clinically lead to positive sensory symptoms (eg, allodynia,

aftersensations, enhanced temporal summation) and negative sensory symptoms (eg, hypoesthesia).[11 16–19] Because mechanisms related to central sensitisation can currently only be assessed in animals,[11] proxies are used in humans to assess clinical signs that may be associated with the presence of central pain mechanisms. We use the term 'human assumed central sensitisation' (HACS) because there is no gold standard to estimate the presence of central sensitisation in humans at present. Moreover, there is also no clear definition, method or (clinical) guideline applicable to diagnose central sensitisation in humans.

The role of HACS in the development of CLBPr, in the chronification of pain, and the interaction with diagnostic and therapeutical interventions is not clear.[11] Several methods are thought to assist in the clinical diagnosis of HACS.[11] With Quantitative Sensory Testing (QST), we measure the responses to calibrated graded innocuous or noxious stimuli[20] and evaluate the presence of HACS in patients with CLBPr.[21] Mehta *et al* showed that following pRF treatment patients with CLBPr have increased pressure pain thresholds and increased conditioned pain modulation, suggesting a normalisation of the sensory pain profile.[21] The presence of HACS might also be evaluated with the Central Sensitisation Inventory (CSI),[22] but there are no clinical associations established between the CSI and experimental measures.[11]

## Objectives

The primary objective of this study is to determine if therapeutic selective nerve root block or pulsed radiofrequency in patients with chronic low back pain radiating to one leg changes the outcome of the CSI and the QST measures as proxies for HACS. Secondary objectives are (1) to determine if HACS can be assessed using the outcome of the CSI in patients with chronic low back pain radiating to one leg, (2) to determine if a dSNRB changes the outcome of the CSI and the QST measures as proxies for HACS in patients with chronic low back pain radiating to one leg, and (3) to determine whether CSI and QST measures as proxies for HACS (both pretreatment and post-treatment) among patients differ from those among sex-matched and aged-matched healthy volunteers.

## Trial design

A prospective longitudinal observational study about HACS in patients with CLBPr who undergo selective nerve root blocks compared with sex-matched and age-matched healthy volunteers.

## METHODS

This study was registered prospectively in the Netherlands Trial Register: Trial NL6765 (first posted 10 January 2018). The trial information in this protocol is in line with the WHO Trial Registration Data Set[23] and written following the SPIRIT 2013 Guideline.[24]

## Participants, interventions and outcomes
### Study setting

Patient recruitment, treatment and measurements take place in the multidisciplinary academic pain center, University Medical Center Groningen (UMCG), Groningen, the Netherlands.

### Patient and public involvement

Patients were not involved in the development of the research question and the determination of the outcome measures. During recruitment, patients are asked to suggest healthy volunteers whose age and gender match their own to participate in the study.

### Eligibility criteria

Fifty consecutive adult patients of both sexes with chronic low back pain radiating in the dermatomes L3 to S2 and 50 sex-matched and age-matched healthy volunteers will be included. The inclusion and exclusion criteria are summarised in table 1. The inclusion criteria for the healthy volunteers are shown in table 2.

Patients can retain their ongoing pain medication or discontinue it if they stop feeling pain during the study. Other changes in medication use are not allowed during participation. Moreover, all other pain interventions (eg, physical therapy, transcutaneous electrical nerve stimulation) during their participation in this study are prohibited. Healthy volunteers are matched with the patients on sex and age (plus or minus 3 years of age). As mentioned, patients are asked to suggest suitable healthy volunteers. If necessary healthy volunteers are also recruited via flyers

**Table 1** Criteria for the inclusion and exclusion of patients

| Inclusion criteria | Exclusion criteria |
|---|---|
| ► Male and female patients<br>► Age 18 years or older<br>► Presence of chronic low back pain radiating in the leg<br>► Leg pain more or equal to back pain<br>► A physician must consider therapeutic sensory nerve root blocking or pulsed radiofrequency as an appropriate treatment intervention<br>► Agreement and signature of the informed consent | ► Exclusion criteria for selective nerve blocks, according to local protocol<br>► Not or not sufficient understanding of the Dutch language<br>► Incapacity to follow instructions<br>► Mental incompetence to provide informed consent<br>► Chronic low back pain with radiation to both legs<br>► Pain in one (or more) sites where quantitative sensory testing will be applied except for the most painful point in the painful dermatome |

**Table 2** Criteria for the inclusion and exclusion of healthy volunteers

| Inclusion criteria | Exclusion criteria |
|---|---|
| ► Male and female healthy volunteers<br>► Age 18 years or older<br>► No history of low back pain<br>► Agreement and signature of the consent | ► Not matching with one of the included patients based on sex and age (plus or minus 3 years of age)<br>► Not or not sufficient understanding of the Dutch language<br>► Incapacity to follow instructions<br>► Mental incompetence to provide informed consent |

placed near to the UMCG facilities. Healthy volunteers do not undergo selective nerve root blocks. Patients and healthy volunteers may withdraw from the study at any time for any reason if they wish to do so and without any consequences. The investigator can decide to withdraw a participant from the study for urgent medical reasons.

## Interventions

The intervention provided in this study is performed in the same way as the standard intervention provided in the multidisciplinary academic pain centre and follows the guideline for SNRB (see online supplemental appendix B for the detailed procedure). Table 3 shows which intervention is used in each visit.

## Participant timeline

Patients and healthy volunteers receive a verbal and written explanation of the study procedures and are given sufficient time to consider their participation. After written informed consent is provided, a meeting is scheduled. The flowchart of the study is shown in figure 1. The schedule for the patient and the healthy volunteer

**Table 3** Overview of the variables collected and interventions per visit

| | | Patients | | | | Healthy volunteers |
|---|---|---|---|---|---|---|
| Demographics | Age, gender, weight, height, comorbidities, pain medication, ethnic background and education. | | | | | |
| | | V1a | V1b, V1c | V2 | V3 | HV |
| Time between visits | | – | 1 week | 1 week | 4 weeks | – |
| Questionnaires | CSI | ✓ | ✓ | ✓ | ✓ | ✓ |
| | SBST | ✓ | – | – | ✓ | ✓ |
| | RAND-36 | ✓ | – | – | ✓ | ✓ |
| | PDI | ✓ | – | – | ✓ | ✓ |
| | WAI | ✓ | – | – | ✓ | ✓ |
| | PVAQ | ✓ | – | – | ✓ | ✓ |
| | PCS | ✓ | – | – | ✓ | ✓ |
| | Pain drawing | ✓ | – | – | ✓ | – |
| | NRS pain | ✓ | ✓ | ✓ | ✓ | ✓ |
| QST | MDT | ✓ | ✓ | ✓ | ✓ | ✓ |
| | DMA | ✓ | ✓ | ✓ | ✓ | ✓ |
| | MPT | ✓ | ✓ | ✓ | ✓ | ✓ |
| | WUR | ✓ | ✓ | ✓ | ✓ | ✓ |
| | PPT | ✓ | ✓ | ✓ | ✓ | ✓ |
| | Cuff PPT | ✓ | ✓ | ✓ | ✓ | ✓ |
| | Cuff TS | ✓ | ✓ | ✓ | ✓ | ✓ |
| | CPM | ✓ | ✓ | ✓ | ✓ | ✓ |
| Intervention | dSNRB | ✓ | ✓ | – | – | – |
| | tSNRB or pRF | – | – | ✓ | – | – |

CPM, conditioned pain modulation; CSI, Central Sensitisation Inventory; Cuff PPT, pressure pain threshold by cuff algometer; Cuff TS, temporal summation by cuff algometer; DMA, dynamic mechanical allodynia; dSNRB, diagnostic selective nerve root block; HV, healthy volunteers; MDT, mechanical detection threshold; MPT, mechanical pain threshold; NRS, Numerical Rating Scale; PCS, Pain Catastophising Scale; PDI, Pain Disability Index; PPT, pressure pain threshold; pRF, pulsed radiofrequency; PVAQ, Pain Vigilance and Awareness Questionnaire; QST, Quantitative Sensory Testing; RAND-36, RAND 36-Item Health Survey; SBST, STarT Back Screening Tool; tSNRB, therapeutic selective nerve root block; V, visit; WAI, Work Ability Index; WUR, wind-up ratio.

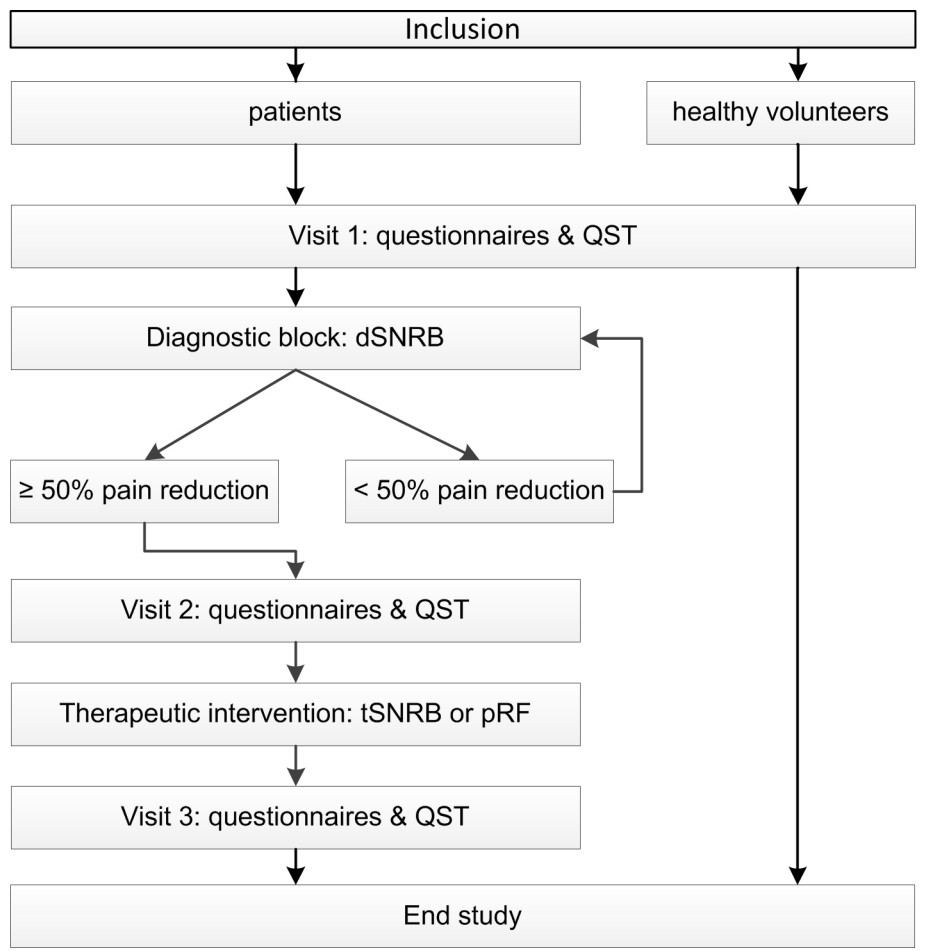

**Figure 1** Flow chart of the study procedure. dSNRB, diagnostic selective nerve root block; pRF, pulsed radiofrequency; QST, quantitative sensory testing; tSNRB: therapeutic selective nerve root block.

assessments are shown in table 3. Visit 1b, and 1c are only performed for the patients when more than one dSNRB is necessary. After the dSNRB, Numerical Rating Scale (NRS) for pain in the back as well as in the leg is assessed. In case of a pain reduction of ≥50%, the patient is considered to have a 'positive result', and the therapeutic intervention is planned. The responsible physician will decide which intervention (tSNRB or pRF treatment) each patient will receive (which is part of care as usual). If a pain reduction is <50%, the patient is considered to have a 'negative result'. With a negative result, the patient will undergo another dSNRB at a different spinal level (three dSNRBs are the maximum within this study). dSRBs, tSNRBs and pRF will be performed according to the standard procedures of the multidisciplinary academic pain centre. Healthy volunteers are not asked to draw their pain area and do not receive any diagnostic or therapeutic intervention. For healthy volunteers, only baseline assessment is performed.

### Recruitment

At the multidisciplinary academic pain centre, on average, two new patients with CLBPr are seen per week. If two patients per month agree to join the study, the sample will be completed within 4 years. Because

patients must visit the multidisciplinary academic pain centre once more than during care as usual (visit 3), they will be compensated for this visit when it is not possible to schedule it with another regular hospital visit. Patients will receive compensation for extra travel costs (€0.19/km). Healthy volunteers will receive 10 euros for their visit plus compensation for travel costs (€0.19/km).

### Duration

Each visit will last no longer than two and a half hours, consisting of 1-hour measurements (patients and healthy volunteers), 0.5 hour for the treatment and 0.5–1 hour for the assessment of the effect of the treatment (patients). Interventions are performed 1 week apart, and visit 3 will be 4 weeks after the therapeutic intervention. Depending on the number of dSNRBs performed, the total duration of this study for the patient will take between 5 and 7 weeks (3–5 visits). Healthy volunteers will undergo only one visit with an assessment of 1 hour.

The first patient was included on 5 June 2018. The planned end date for the inclusion of participants is 1 March 2022.

## Outcomes

All measurements are performed for study-specific reasons. The primary outcomes are QST measures and the CSI. Secondary outcomes are Pain Catastrophising Scale, Pain Vigilance and Awareness Questionnaire, 36-Item Short-Form Health Survey, Pain Disability Index, Work Ability Index, STarT Back Screening Tool (SBST), NRS and drawing in standard leg images (pain drawing) (table 3).

## Descriptive data

Descriptive items include: age (years), sex (male / female), weight (kg), height (cm), education, comorbidities, medication and ethnic background.

## Questionnaires

1 or 2 days before each visit, the patient receives an email with a link and is asked to fill out the questionnaires (table 3). The questionnaires (provided in Dutch) are sent through RoQua (UMCG, Groningen, The Netherlands), a questionnaire programme built into the electronic patient's file. The whole set of questionnaires is sent for visit 1a and 3, and it takes about 20 min to complete. For visit 1b-1c and 2, only the CSI is sent, and take about 5 min to complete. Patients who cannot fill out the questionnaires online will receive them on paper at the beginning of the visit.

*CSI:*[22 25] the CSI contains 25 statements related to current health symptoms. Each item is measured on a 5-point Likert scale from 0 (never) to 4 (always). A cut-off score of 40 is being used.[22 25] The original CSI (Cronbach's alpha=0.879, test–retest reliability=0.817)[25] as well as the CSI-Dutch language version (Dlv) (Cronbach's alpha=0.91, test–retest reliability=0.88 and 0.91)[22] shows a good internal consistency.

*Pain Catastrophizing Scale:*[26] it assesses thoughts and feelings about pain. For each of 13 statements, the participant is asked to answer on a scale from 0 (totally not) to 4 (always). The original pain catastrophising scale (Cronbach's alpha=0.93,[27] test–retest reliability=0.75)[26] as well as the Pain Catastrophizing Scale-Dlv (Cronbach's alpha between 0.85 and 0.91)[28–30] show good internal consistency.

*Pain Vigilance and Awareness Questionnaire:*[31 32] this questionnaire evaluates the awareness of pain. The participant can assign a number from 0 (never) to 5 (continuously) for 16 statements. The original pain vigilance and awareness questionnaire shows good internal consistency (Cronbach's alpha=0.86)[31] and adequate test–retest reliability (r=0.80).[31] The Pain Vigilance and Awareness Questionnaire-Dlv also shows internal consistency (Cronbach's alpha=0.87).[32]

*RAND 36-item Health Survey (RAND-36):*[33 34] the RAND-36 is a participant-reported survey of health-related quality of life. It consists of 8 sections: vitality, physical functioning, bodily pain, general health perceptions, physical role functioning, emotional role functioning, social role functioning and mental health. The RAND-36-Dlv has shown to be valid and reliable.[33]

*Pain Disability Index:*[35 36] this questionnaire is used to measure the degree to which aspects of the patient's life is disrupted by chronic pain. It lists 7 life activity categories (family/home responsibilities, recreation, social activity, occupation, sexual behaviour, self-care and life-support activities). The original pain disability index shows good internal consistency (Cronbach's alpha=0.871),[37] and the test–retest reliability is good (ICC=0.91).[38] The Pain Disability Index-Dlv showed good internal consistency with patients with chronic low back pain (Cronbach's alpha=0.85)[36] and sufficient test–retest reliability (ICC=0.78).[36]

*Work Ability Index:*[39] it is used to assess workability during health examinations. The index is determined based on the answers to a series of questions that consider the demands of work, the worker's health status and resources. The original work ability index has been validated (Cronbach's alpha=0.72).[40 41] The Work Ability Index-Dlv showed sufficient internal consistency (Cronbach's alpha=0.72)[41] and an acceptable test–retest reliability.[42]

*SBST:*[43 44] the SBST is a 9-item patient self-reporting questionnaire validated for triage of patients with non-specific low back pain in primary care. It has shown to be helpful in tertiary care.[45] The SBST identifies modifiable prognostic factors from the health domains of pain, activity limitation and psychosocial factors, which are risk factors for persistent non-specific low back pain. The SBST classifies patients into three groups: low, medium or high risk of poor prognosis based on the symptom complexity. When assessed by general practitioners, this classification can assist in making decisions towards appropriate evidence-based treatment pathways. The SBST-Dlv showed good validity and an excellent to fair reproducibility.[44]

*NRS for Pain:* a pain rating scale where patients are asked to rate their pain at two locations (leg and back), giving a number between 0 and 10 (0=no pain/10=maximum pain imaginable). In the QST measurements, an NRS is also used with a scale from 0 to 100 (0=no pain/100=maximum pain imaginable).

*Drawing in standard leg images (Pain Drawing):* the patient is asked to draw, on an image with standard legs (see online supplemental appendix C), the area where they experience the most pain. When more locations are drawn, an arrow will indicate the most painful spot.

## Quantitative Sensory Testing

QST is a psychophysical method that measures responses to calibrated graded innocuous or noxious stimuli in addition to bedside clinical examination of the somatosensory system.[19 46] The measurements are based on the QST battery developed by the German Research Network on Neuropathic Pain.[10 15] This protocol is extended with cuff algometry (NociTech, Aalborg, Denmark). QST is used for the analyses of HACS. The participants will be

**Table 4** Sequence of Quantitative Sensory Testing measurement sites

| Visit | Sequence |
| --- | --- |
| Visit 1a | Site D-A-C-B |
| Visit 1b | Site D-C-A-B |
| Visit 1c | Site D-B-A-C |
| Visit 2 | Site D-C-B-A |
| Visit 3 | Site D-B-C-A |

comfortable sitting and lying on a bed during all assessments when evaluating the stimuli. The sequences' order is counterbalanced and documented in the case report form (table 4). The measurements take about 45–60 min.

*Mechanical detection threshold:* mechanical detection threshold is assessed via von Frey-filaments (OptiHair2, MRC Systems GmbH, Heidelberg, Germany). When the 16 mN von Frey filament's touch is felt, the von Frey filament's intensity is reduced until no touch is felt. The intensity then will be increased until the touch is felt again. This test will be repeated at least three times to detect the mechanical detection threshold. The first von Frey filament that cannot be felt is documented.[47–49]

*Dynamic mechanical allodynia:* dynamic mechanical allodynia is tested using dynamic innocuous stimulus (soft brush; Sense Lab Brush no. 5, Somedic, Sösdala, Sweden). The innocuous stimulus is administered once and after that three times in a 1–2 cm wiping motion on the skin. The participant is asked to say if something was felt and to rate the NRS-Pain from 0 to 100.[50–52]

*Mechanical pain threshold:* this test is performed with weighted pinprick stimuli (MRC Systems GmbH, Heidelberg, Germany) using the method of limits. Five threshold determinations are made, each with a series of ascending and descending stimulus intensities. The first pinprick that is considered painful is documented.[51 53–55]

*Wind-up ratio:* in this test, the NRS-Pain score (0–100) experienced for a single pinprick stimuli (256N) is compared with an applied series of repetitive (10 times with intervals of 1 s) pinprick stimuli of the same intensity (256N). The wind-up ratio is calculated by dividing the pain intensity rating for the series of stimuli by the pain intensity rating for the single stimulus.

*Pressure pain threshold:* using a handheld pressure algometer (Wagner FDX 10, Greenwich, USA), the threshold for pressure-induced pain is assessed by slowly increasing stimulus intensities (5 N/s). The threshold is determined when the pressure becomes painful/more than just the feeling of pressure when the participants say 'now'. The NRS-Pain (0–100) and the applied force (N) at that moment are documented.

*Cuff algometry (Cuff):* during cuff algometry, the participants will be instructed to continuously evaluate the pain using an electronic visual analogue scale (0=no pain, 10=worst imaginable pain).

*Cuff pain tolerance threshold (PTT):* the cuff pressure will be increased by 1 kPa/s on the dominant leg. The participant will be instructed to rate the pain intensity continuously on the electronic visual analogue scale until the tolerance level is reached, and the participant pushes the stop button. When the participant pushes this button, the pressure is released immediately. The PTT is defined as when the participant presses the stop button. A similar assessment will be performed on the non-dominant leg after the cuff temporal summation.

*Cuff temporal summation:* a total of 10 repeated mechanical cuff pressure stimuli at the intensity of PTT will be delivered at 0.5 Hz (1s stimulus duration and 1s interval between stimuli) to the lower leg. A constant pressure of 1 kPa will be applied between the individual pressure stimuli sets to avoid movement of the cuff. The participants will continuously rate the pain intensity on the electronic visual analogue scale during the 10 repeated stimuli.

*Conditioned pain modulation:* pressure pain threshold will be repeated on the deltoid muscle and the rectus femoris, both on the painful dermatome's contralateral side. The participant will be asked to put the dominant hand up to the wrist with spread fingers in a bucket filled with ice and cold water (conditioning stimulus). The amount of time (at a maximum of 3 min for safety reasons) that it takes until the participant removes the immersed hand out of the bucket will be noted. Immediately after, pressure pain threshold in the deltoid muscle and the rectus femoris both on the contralateral side of the painful dermatome will be repeated.[17 18 20]

*Measurement sites:* measures will be taken at 6 different sites in total (table 5). The QST measures will be performed on sites A–D. The conditioned pain modulation will be assessed at sites D and E. The cuff measures will be performed at site F.

## Sample size

The sample size was calculated via G*power for Windows (V.3.1.9.7, Heinrich Heine Universität, Dusseldorf, Germany) and was calculated to be 41 individuals per group. As a safety margin for possible correction for dropout or missing data, we added 20% to result in 50 patients and 50 healthy volunteers.

The required sample size to answer the first objective was based on the difference between two dependent means (matched pairs). Mehta *et al*[21] described an increase in PPT from 310±90 towards 375±90. The calculated effect size (d) was 0.72 and with a power of 90% and a type I error of 5% the calculated number of patients to be included was 22.

For objective 2, the sample size calculation was based on the Wilcoxon-Mann-Whitney test for two groups (patients and healthy volunteers). The number of participants to compare the patients with the healthy volunteers was calculated based on the paper by Blumenstiel *et al*.[56] The PPT threshold in 23 patients with chronic back pain (mean 239.3 kPa; 95% CI 200 to 287) compared with 20

| Table 5 | Quantitative sensory testing measurement sites |
|---|---|
| Site A | The most painful point in the painful dermatome |
| Site B | Contralateral point of location A |
| Site C | Control site 1 (contralateral of location A, distant from the painful area), between the scapulae. On the medial part of trapezius muscle at the height of the spina scapulae, 4 cm lateral to the spinous process of the third thoracic vertebra (Th3) |
| Site D | Control site 2 (distant from the painful area): contralateral to location A, at the deltoid muscle: on the medial part of the deltoid muscle, 3 cm below the acromion |
| Site E | On the rectus femoris muscle 15 cm proximal to the base of the patella |
| Site F | On the lower leg. At the level of the largest circumference of gastrocnemius muscle. |

healthy controls (mean 352 kpa; 95% CI 286 to 432). The calculated effect size (d) was 0.75, and with a power of 90% and a type I error of 5% the calculated number of patients and volunteers was in total 82 (41 individuals per group).

## Data collection, management and analysis
### Data collection and management
The primary data (raw) will be collected on paper and entered into an electronic data capture: OpenClinica (OpenClinica, LLC, Waltham, USA, V.3.14). The questionnaires will be assessed via RoQua (RoQua, UMCG, Groningen, The Netherlands) as implemented in the electronic patient files (EPIC, Epic Systems Corporation, Verona, USA, version August 2020). For the healthy volunteers' dummy numbers (for the hospital information system) will be created to enable them to access and fill in the questionnaires. The data in Open Clinica and RoQua will be combined into SPSS (V.27, IBM) database.

The raw data on paper, in OpenClinica and the data of RoQua, will be anonymised and coded. The data will be stored at the UMCG research drive.

Data will be handled confidentially. A participant identification code list will be used to link the data to the participant. The principal investigator will safeguard the key to the code.

Data will be accessible for the team of investigators, the Medical Ethical Review Board and the healthcare Inspection. Data will be saved for 15 years. Data will be used for publication, but no participant will be traceable. The handling of personal data will comply with the Dutch Personal Data Protection Act.

## Statistical methods
Data records from the different tests and questionnaires will be collected and merged into one database. SPSS software V.27.0 or higher (IBM) will be used to perform the analyses. Descriptive statistics data will be presented as means±SD when normally distributed and as median and IQR (25–75) when not normally distributed. A significance level of 0.05 is used in all analyses.

For the primary objective, to determine treatment effect of the tSNRB and pRF the difference between visit 1a and visit 3 will be analysed for 9 primary measures (CSI and 8 QST measures). This analysis will be performed using a student's paired t-test in normally distributed data or a Wilcoxon test in non-normally distributed data.

For secondary objective 1, to determine if the presence of HACS in patients with CLBPr can be assessed using the CSI, the data from visit 1a will be used.

For secondary objective 2, the difference between the latest visit 1 and visit 2 will be used to determine treatment effect of the dSNRB. The 9 primary measures (CSI and 8 QST measures) will be used. The difference will be assessed using a student's paired t-test or a Wilcoxon test where appropriate.

For secondary objective 3, the differences between patients' visit 1a, versus visit 3 and healthy volunteers are calculated for the questionnaires and QST measurements. The calculated differences (patients' visit 1a vs healthy volunteers and patients' visit 3 vs healthy volunteers) will be assessed using an a student t-test for normally distributed data and a Mann-Whitney U test for non-normally distributed data

## MONITORING
### Data monitoring
The monitoring will be carried out by research support staff based on GCP monitoring principles. The departmental research office will monitor the trial regularly. This typically includes the following checks:

(1) The data collected are consistent with adherence to the trial protocol; (2) case report files are completed by authorised persons; (3) no key data are missing; (4) the data appear to be valid (ie, range and outlier checks); (5) review of recruitment rates, withdrawals and losses to follow-up.

### Harms
Experienced specialised pain staff members perform all interventions in the care as usual setting. Study-related additional harms may consist of QST-related pain during and a short time after the measurements. No other harm is expected.

### Protocol amendments
The following amendments were made before the inclusion of the fifth patient. Regarding the inclusion criteria, the age limit of 65 was excluded. The visual analogue scale was changed to NRS because NRS is used in care as usual. Before the first healthy volunteer's inclusion, the number of visits was reduced from 3 to 1. The latest protocol, version 5, is from 6 July 2020. Due to the COVID-19 pandemic, the study was put on hold, and an amendment (version 5) was submitted and approved on 11 August 2020, to prolong the initial inclusion date for the participants to 1 March 2022.

### Declaration of interest

Not applicable.

### IPD sharing statement

The deidentified individual clinical trial participant-level data (IPD) will be shared. All individual participant data that underlie the results reported in this article will be shared after deidentification (text, tables, figures and appendices). The study protocol will also be available. The data will be available beginning 9 months following article publication and for a maximum period of 15 years. To make the data findable and accessible for others, we will include a description of the UMCG data catalogue data: https://wwwgroningendatacatalogusnl/. Researchers who provide a methodologically sound proposal can access the data with a signed data access agreement.

**Contributors** HT, GJG, KKP, LA-N and APW designed the trial protocol. GJG and APW secured funding for the study. IS, HT and APW drafted the manuscript. GJG, LA-N and KKP contributed to the manuscript. All authors read and approved the final manuscript. IS, HT and APW are involved in data collection during the study. All authors are involved in the data analysis and manuscript preparations after the study is completed.

**Funding** The University of Groningen provided funding, University Medical Center Groningen, the Graduate Sçhool of Medicine and Department of Anesthesiology, Pain Center, Groningen, the Netherlands and the Ministry of Education of Brazil—Science without Borders programme (PhD scholarship to Ms Tharcila Chaves—project number 9140/13-1), Rio de Janeiro, Brazil.

**Disclaimer** The funding bodies had no role in the study design and collection, analysis and interpretation of data, and writing of the manuscript.

**Competing interests** None declared.

**Patient consent for publication** Not applicable.

**Provenance and peer review** Not commissioned; externally peer reviewed.

**ORCID iDs**

Ingrid Schuttert http://orcid.org/0000-0003-2164-4721
Hans Timmerman http://orcid.org/0000-0002-6082-9043
Gerbrand J Groen http://orcid.org/0000-0002-7199-0107
Kristian Kjær Petersen http://orcid.org/0000-0003-4506-000X
Lars Arendt-Nielsen http://orcid.org/0000-0003-0892-1579
Andre P Wolff http://orcid.org/0000-0001-6240-3903

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
