## [Reviewer comments · BMJ Open]

ARTICLE DETAILS

TITLE (PROVISIONAL)	Human Assumed Central Sensitisation (HACS) in Patients with Chronic Low Back Pain Radiating to the Leg (CLaSSICO study):Protocol for a Prospective Observational Study.
AUTHORS	Schutttert, Ingrid; Timmerman, Hans; Groen, Gerbrand; Petersen, Kristian; Arendt-Nielsen, Lars; Wolff, Andre

VERSION 1 – REVIEW

REVIEWER	Rosie Cornish University of Bristol, School of Social and Community Medicine
REVIEW RETURNED	16-Jun-2021

GENERAL COMMENTS	Overall, I found the protocol slightly difficult to follow, perhaps because of all the acronyms, the fact that the outcome (HACS) is actually quantified using a number of different measures (CSI, QST etc) and the fact that – in my opinion - the objectives are not very clearly stated. 1. To address this last point, it might be helpful to firstly write down what the research questions are, then translate these to clearly stated objectives (which should then each map to a specific analysis – see below). It seems clear what some of these are: Do tSNRB and pRF give rise to changes in HACS? How does HACS change after dSNRB? How does HACS compare in patients versus healthy volunteers? But I think one objective is to determine how best to measure HACS? (I am not sure whether this is correct) 2. It would also be helpful to mention the outcome measures in the objectives section and to ensure that the statistical analysis section matches the objectives. The primary objective of the study is stated as being “to determine HACS changes caused by therapeutic selective nerve root block or pulsed radiofrequency...”. The statistical analysis states that this will be assessed using ANOVA with factors group, side, and site. The outcomes section states that there are 8 primary outcomes (CSI and the seven QST measures) and 8 secondary outcomes (PCS etc). Thus, the objective to me is to compare post treatment (where applicable) levels of HACS as measured by 8 primary and 8 secondary outcomes in the three groups (tSNRB, pRF, and healthy volunteers). Or is the plan to also compare pre-treatment levels? (If not, this comparison is also important since presumably the two treatment groups may not necessarily be comparable at baseline?). The secondary objectives are listed as: (1) to assess HACS in patients, (2) to determine change in HACS caused by dSNRB, (3) to compare differences in HACS between patients (before and after SNRB) and healthy volunteers. For (1) the analysis plan is to use correlation coefficients and then to do repeated measures ANOVA to assess changes in HACS. I could not work out whether the purpose of this analysis was
---

	to determine which of the 16 outcome measures were good measures of HACS or just to describe how these outcomes were related and how they changed over time – following SNRB. It would be helpful to state this more explicitly. For (2) the analysis does not seem to match up with the objective. The analysis is an ANOVA (presumably on measures taken at visit 2 for patients and measures taken at visit 1 for healthy volunteers). This does not assess change in HACS; it will just determine whether outcomes after dSNRB in patients differ from (normal) outcomes in healthy volunteers. To assess change, the before and after measures need to be compared. Finally, for (3) the analysis states that differences (baseline vs follow-up) and healthy volunteers (baseline only) will be compared using an unpaired t-test. You cannot use an unpaired t-test to compare two different outcomes (i.e. differences: follow-up minus baseline in the intervention groups and baseline levels in the volunteer group). As above, to assess differences before and after treatment, you can only compare the two groups where differences have been measured.
--	---

VERSION 1 – AUTHOR RESPONSE

Reviewer Report 1: Dr. Rosie Cornish, University of Bristol

Dear Dr. Cornish,

Thank you for your helpful feedback and insights. Based on your suggestions and comments, we made the following changes to the manuscript.

Comments to the Author:

Overall, I found the protocol slightly difficult to follow, perhaps because of all the acronyms, the fact that the outcome (HACS) is actually quantified using a number of different measures (CSI, QST etc) and the fact that – in my opinion - the objectives are not very clearly stated.

Reply: To increase the readability of the manuscript, we have changed the abbreviations into full names when they were used less than five times in the manuscript text (page 4-6: the abbreviations list). However, we use different measures to assess (human assumed) central sensitisation because there is no gold standard.. We chose to include these multiple techniques for the identification of HACS (the Central Sensitisation Inventory and several Quantitative Sensory Testing techniques) because these measures are most frequently used in other studies.

The objectives are – in my opinion - not very clearly stated. To address this last point, it might be helpful to firstly write down what the research questions are, then translate these to clearly stated

objectives (which should then each map to a specific analysis – see below). It seems clear what some of these are: Do tSNRB and pRF give rise to changes in HACS? How does HACS change after dSNRB? How does HACS compare in patients versus healthy volunteers? But I think one objective is to determine how best to measure HACS? (I am not sure whether this is correct)

Reply: Thank you for your guidance. The primary objective is rewritten based on your remarks into "to determine if therapeutic selective nerve root block or pulsed radiofrequency in patients with chronic low back pain radiating to one leg changes the outcome of the CSI and the QST measures as proxies for human assumed central sensitisation." (page 10)

The secondary objectives were also changed based on your remarks::

(1) to identify human assumed central sensitisation based on the outcome of the CSI in patients with chronic low back pain radiating to one leg

(2) to determine if a diagnostic selective nerve root block changes the outcome of the CSI and the QST measures as proxies for human assumed central sensitisation in patients with chronic low back pain radiating to one leg

(3) to compare the differences in the outcome of the CSI and QST measures as proxies for human assumed central sensitisation between patients (before and after selective nerve root blocks) and matched healthy volunteers.

It would also be helpful to mention the outcome measures in the objectives section and to ensure that the statistical analysis section matches the objectives.

The primary objective of the study is stated as being "to determine HACS changes caused by therapeutic selective nerve root block or pulsed radiofrequency...". The statistical analysis states that this will be assessed using ANOVA with factors group, side, and site. The outcomes section states that there are 8 primary outcomes (CSI and the seven QST measures) and 8 secondary outcomes (PCS etc). Thus, the objective to me is to compare post treatment (where applicable) levels of HACS as measured by 8 primary and 8 secondary outcomes in the three groups (tSNRB, pRF, and healthy volunteers). Or is the plan to also compare pre-treatment levels? (If not, this comparison is also important since presumably the two treatment groups may not necessarily be comparable at baseline?).

The secondary objectives are listed as: (1) to assess HACS in patients, (2) to determine change in HACS caused by dSNRB, (3) to compare differences in HACS between patients (before and after SNRB) and healthy volunteers. For (1) the analysis plan is to use correlation coefficients

and then to do repeated measures ANOVA to assess changes in HACS. I could not work out whether the purpose of this analysis was to determine which of the 16 outcome measures were good measures of HACS or just to describe how these outcomes were related and how they changed over time – following SNRB. It would be helpful to state this more explicitly.

For (2) the analysis does not seem to match up with the objective. The analysis is an ANOVA (presumably on measures taken at visit 2 for patients and measures taken at visit 1 for healthy volunteers). This does not assess change in HACS; it will just determine whether outcomes after dSNRB in patients differ from (normal) outcomes in healthy volunteers. To assess change, the before and after measures need to be compared.

Finally, for (3) the analysis states that differences (baseline vs follow-up) and healthy volunteers (baseline only) will be compared using an unpaired t-test. You cannot use an unpaired t-test to compare two different outcomes (i.e. differences: follow-up minus baseline in the intervention groups and baseline levels in the volunteer group). As above, to assess differences before and after treatment, you can only compare the two groups where differences have been measured.

Reply: Thank you for your advice on the statistics section. We revised this section following your comments and focused on each objective separately (page 19).

For the first objective, we stated to perform an ANOVA for group, side and site. This is changed into: *'to determine treatment effect of the tSNRB and pRF the difference between visit 1a and visit 3 will be analysed for nine primary measures (CSI and eight QST measures). The results will be checked for possible covariates. This analysis will be performed using an One-way ANOVA.'*

For the second objective, part 1, we stated to use Pearson's correlation between the different HACS indicators and use a repeated-measures ANOVA to compare values before and after the intervention. The CSI cut-off value of 40 will be used to identify HACS for this objective; therefore, we changed this part to: *'to identify the presence of HACS in patients with CLBPr data from visit 1a will be used. The CSI cut-off value of 40 will be used to identify the presence of HACS.'*

For the secondary objective, part 2, we stated to perform an ANOVA for group, side and site. This was indeed vaguely formulated; therefore, we changed the objective to: *'the difference between the latest visit 1 and visit 2 will be used to determine treatment effect of the dSNRB. The nine primary measures (CSI and eight QST measures) will be used and checked for*

possible covariates for this analysis. The difference will be assessed using an One-way ANOVA.'

For the secondary objective, part 3, we used incorrect analyses. Therefore this is changed into *'For secondary objective 3, the differences between patients visit 1a vs visit 3 and healthy volunteers are calculated for the questionnaires and QST measurements. The calculated differences will be assessed using an one-way ANOVA.'*

VERSION 2 – REVIEW

REVIEWER	Rosie Cornish University of Bristol, School of Social and Community Medicine
REVIEW RETURNED	01-Nov-2021

GENERAL COMMENTS	1. I am still having trouble seeing how the statistical analysis matches up with the objectives. Specific points are given below. a. The primary objective and secondary objective 2 are now clearer to me but I cannot see how a one-way ANOVA is appropriate for either of these. The question is: “Is there a difference between baseline and post-treatment values among those treated?” These are paired data. One way ANOVA is used to determine whether there is a difference, on average, between three or more independent groups. And what is meant by, “The results will be checked for possible covariates”? Do the authors want to know whether the effect of treatment is different in different subgroups? b. Secondary objective 1 is still unclear. The objective is stated as being to identify HACS using CSI. I don’t quite know what the research question is here – is it: “Can CSI be used to identify HACS?” If yes, then how does the analysis (using a cut-off of 40) answer this question? If the question is something different from this, then this needs to be stated more clearly. c. My comment about secondary objective 3 is very similar to what I initially stated: The analysis states that differences between patients’ (visit 1a vs visit 3) and healthy volunteers will be calculated and compared using one-way ANOVA. Unless I have misunderstood again, I think this means the authors will be calculating a difference for patients (visit 3 minus visit 1a) for each outcome and comparing this to baseline values in the volunteer group? As I said previously, these cannot be compared because they are different outcomes. Or perhaps the word “differences” is confusing me? Are they going to compare baseline values – patients versus volunteers – then separately compare post-treatment values - patients versus volunteers? If so, maybe the objective needs rewording as: To determine whether CSI and QST measures (both pre-treatment and post-treatment) among patients differ from those among healthy volunteers. 2. The lack of clarity in the analysis makes me question how the sample size was calculated. I presume it was done on the basis of using an unpaired t-test to compare patients to healthy volunteers? Was this on the basis of pre- or post-treatment values of the QST measures?
---

VERSION 2 – AUTHOR RESPONSE

Reviewer Report 1: Dr. Rosie Cornish, University of Bristol

Dear Dr. Cornish,

Thank you for your helpful feedback and insights. Based on your suggestions and comments, we had a meeting/discussion with a statistician of our epidemiology department. As a result we made the following changes to the manuscript.

Comments to the Author:

1. I am still having trouble seeing how the statistical analysis matches up with the objectives. Specific points are given below.

a. The primary objective and secondary objective 2 are now clearer to me but I cannot see how a one-way ANOVA is appropriate for either of these. The question is: “Is there a difference between baseline and post-treatment values among those treated?” These are paired data. One way ANOVA is used to determine whether there is a difference, on average, between three or more independent groups.

Reply: Indeed it is paired data, therefore a Student’s paired t-test or a Wilcoxon test will be used where appropriate. We rewritten the statistics for the primary objective and added this to the manuscript as: *“This analysis will be performed using a student’s paired t-test in normally distributed data or a Wilcoxon test in non-normally distributed data.”* (page 16 row 416-417)

And what is meant by, “The results will be checked for possible covariates”? Do the authors want to know whether the effect of treatment is different in different subgroups?

Reply: No check for covariates will be performed. Therefore this sentence is deleted. (page 16 row 415)

b. Secondary objective 1 is still unclear. The objective is stated as being to identify HACS using CSI. I don't quite know what the research question is here – is it: “Can CSI be used to identify HACS?” If yes, then how does the analysis (using a cut-off of 40) answer this question? If the question is something different from this, then this needs to be stated more clearly..

Reply: Secondary objective 1 was stated before as “to identify human assumed central sensitisation based on the outcome of the CSI in patients with chronic low back pain radiating to one leg” which is unclear indeed. We have rewritten this objective and added it to the manuscript as: “to determine if human assumed central sensitisation can be assessed using the outcome of the CSI in patients with chronic low back pain radiating to one leg.” (page 8 row 156-157)

The statistics of secondary objective 1 are also changed into: “For secondary objective 1, to identify determine if the presence of HACS in patients with CLBPr data can be assessed using the CSI, the data from visit 1a will be used.” (Page 17 row 418-420)

c. My comment about secondary objective 3 is very similar to what I initially stated: The analysis states that differences between patients' (visit 1a vs visit 3) and healthy volunteers will be calculated and compared using one-way ANOVA. Unless I have misunderstood again, I think this means the authors will be calculating a difference for patients (visit 3 minus visit 1a) for each outcome and comparing this to baseline values in the volunteer group? As I said previously, these cannot be compared because they are different outcomes. Or perhaps the word “differences” is confusing me? Are they going to compare baseline values – patients versus volunteers – then separately compare post-treatment values - patients versus volunteers? If so, maybe the objective needs rewording as: To determine whether CSI and QST measures (both pre-treatment and post-treatment) among patients differ from those among healthy volunteers.

Reply: To create more clarity about secondary objective 3 (stated before as “to compare the differences in the outcome of the CSI and QST measures as proxies for human assumed central sensitisation between patients (before and after selective nerve root blocks) and matched healthy volunteers.”) we have rewritten the objective following your suggestion into “to determine whether CSI and QST measures as proxies for human assumed central sensitisation (both pre-treatment and post-treatment) among patients differ from those among sex and aged matched healthy volunteers” (Page 8 row 161-163)

The statistics of secondary objective 3 are also changed into: “the differences between patients' visit 1a, vs visit 3 and healthy volunteers are calculated for the questionnaires and QST measurements. The calculated differences (patients' visit 1a vs healthy volunteers and patients' visit 3 vs healthy volunteers) will be assessed using an a student t-test for normally distributed data and a Mann-Whitney-U test for non-normally distributed data.” (page 17 row 426-430)

2. The lack of clarity in the analysis makes me question how the sample size was calculated. I presume it was done on the basis of using an unpaired t-test to compare patients to healthy volunteers? Was this on the basis of pre- or post-treatment values of the QST measures?

Reply: Thank you for your feedback. We have checked our sample size calculation. We changed the manuscript regarding the calculation of the sample size into:

*“The sample size was calculated via G*power for Windows (Version 3.1.9.7, Heinrich Heine Universität, Dusseldorf, Germany) and was calculated to be 41 individuals per group. As a safety margin for possible correction for dropout or missing data we added 20% to result in 50 patients and 50 healthy volunteers.*

The required sample size to answer the first objective was based on the difference between two dependent means (matched pairs). Mehta et al²¹ described an increase in PPT from 310 ± 90 towards 375 ± 90. The calculated effect size (d) was 0.72 and with a power of 90% and a type I error of 5% the calculated number of patients to be included was 22.

For objective two the sample size calculation was based on the Wilcoxon-Mann-Whitney test for two groups (patients and healthy volunteers). The number of participants to compare the patients with the healthy volunteers was calculated based on the paper by Blumenstiel et al.⁵⁶. The PPT threshold in 23 patients with chronic back pain (mean 239.3 kPa; 95%CI 200-287) compared to 20 healthy controls (mean 352 kpa; 95%CI 286-432). The calculated effect size (d) was 0.75, and with a power of 90% and a type I error of 5% the calculated number of patients and volunteers was in total 82 (41 individuals per group).” (page 15 row 364-378)

VERSION 3 – REVIEW

REVIEWER	Rosie Cornish University of Bristol, School of Social and Community Medicine
REVIEW RETURNED	09-Dec-2021

GENERAL COMMENTS	The authors have addressed my previous comments. I have no additional ones.
---